# Functional Gastrointestinal Symptoms in Children with Autism and ADHD: Profiles of Hair and Salivary Cortisol, Serum Leptin Concentrations and Externalizing/Internalizing Problems

**DOI:** 10.3390/nu16101538

**Published:** 2024-05-20

**Authors:** Andreas Petropoulos, Sophia Anesiadou, Maria Michou, Aikaterini Lymperatou, Eleftheria Roma, George Chrousos, Panagiota Pervanidou

**Affiliations:** 1Unit of Developmental and Behavioral Pediatrics, First Department of Pediatrics, School of Medicine, National and Kapodistrian University of Athens, “Aghia Sophia” Children’s Hospital, 11527 Athens, Greece; petropoulos_andreas@yahoo.gr (A.P.); sofia.anesiadou@yahoo.gr (S.A.); kliberatou@gmail.com (A.L.); 2School of Medicine, National and Kapodistrian University of Athens, “Aghia Sophia” Children’s Hospital, 11527 Athens, Greece; roma2el@med.uoa.gr (E.R.); chrousge@med.uoa.gr (G.C.); 3Human Ecology Laboratory, Department of Home Economics and Ecology, Harokopio University, 17676 Athens, Greece; mariamixou@hotmail.com; 4Postgraduate Program “The Science of Stress and Stress Promotion”, School of Medicine, National and Kapodistrian University of Athens, “Aghia Sophia” Children’s Hospital, 11527 Athens, Greece

**Keywords:** functional gastrointestinal disorders, neurodevelopmental disorder, autism, HPA axis, cortisol, internalizing and externalizing problems

## Abstract

Background: Functional Gastrointestinal Disorders (FGIDs) present a higher prevalence in individuals with Neurodevelopmental Disorders (NDDs). The Stress System and the Gut–Brain axis (GBA) may mediate these relations. We aimed to assess the prevalence and profile of FGIDs in a clinical sample of children with Autism Spectrum Disorder (ASD) and Attention Deficit/Hyperactivity Disorder (ADHD) compared to typically developing children (TD) as well as to investigate possible relations between stress-related biomarkers and internalizing/externalizing problems in children with NDDS. Methods: In total, 120 children, aged between 4 and 12 years old, formed three groups (N = 40, each): ADHD, ASD and TD. Salivary cortisol, hair cortisol and serum leptin were measured. Results: The ASD group had more FGID problems than the TD group (*p* = 0.001). The ADHD and ASD groups had higher total internalizing/externalizing problems than the TD group (*p* < 0.0001, *p* < 0.0001, *p* = 0.005, respectively). Children with FGIDs showed more total, internalizing and externalizing problems compared to children without FGIDs (*p* < 0.0001, *p* < 0.0001, *p* = 0.041, respectively). The ADHD group showed lower AUCg values (*p* < 0.0001), while the hair cortisol was higher for the TD group (*p* < 0.0001). Conclusion: In conclusion, children with NDDs had more FGID symptoms and present higher internalizing and externalizing problems. Children with ADHD and FGIDs had more internalizing problems compared to those without FGIDs. No differences in stress-related biomarkers were shown to differentiate children with NDDs with and without FGIDs. Future prospective studies including a greater number of children may elucidate the biological pathways linking these comorbidities.

## 1. Introduction

Neurodevelopmental disorders (NDDs) are a complex group of disorders that influence the brain’s function and alter neurobiological development in early childhood [1]. The clinical features of NDDs, according to the Diagnostic and Statistical Manual of Mental Disorders, 5th edition (DSM-5), include impairments in cognition, verbal and non-verbal communication, socialization, behavior and/or fine and gross motor skills, which persist into adulthood [2,3]. Gastrointestinal symptoms are commonly seen in children with NDDs and often occur with internalizing and externalizing comorbidities [4,5]. Among NDDs, Autism Spectrum Disorder (ASD) is characterized by persistent deficits in social communication, restrictive interests, and repetitive behavioral patterns. The prevalence of ASD is increasing worldwide, and it is now estimated in 1 in 36 children in the United States [6,7,8,9]. ASD is frequently accompanied by many physical and mental health comorbidities, including anxiety, depression, stress-related symptoms, and changes in the hypothalamic-pituitary-adrenal axis (HPA) [10,11]. Environmental stressors and sensory deficits may contribute to stress dysregulation [12,13]. Attention-deficit hyperactivity disorder (ADHD) is the most frequent neurodevelopmental disorder, affecting, globally, 5.9–7.2% of children, and it is characterized by symptoms of attention deficit, hyperactivity, and impulsivity, assessed in several settings, [14,15]. Children with ADHD present higher levels of internalizing and externalizing symptoms [16,17,18,19,20].

Functional gastrointestinal disorders (FGIDs) consist of various recurring gastrointestinal (GI) symptoms such as constipation, abdominal pain, vomiting, nausea or diarrhea, which are not identified by any organic or structural factor [21,22,23,24,25]. FGIDs are common in children, with a prevalence of 23% [26]. Several studies have shown that FGIDs may have an increased prevalence in children with NDDs in comparison with the general population. According to Lasheras et al., the prevalence of GI disorders in children with ASD is 13–57%, while other studies ranged them between 9% and 91% [27]. The strong correlation between GI dysfunction and NDDs could be explained through the microbiota–gut–brain axis (MGBA) [28]. The MGBA has been coined to describe a bidirectional communication pathway between the microbiota, gut and central nervous system (CNS), which can explain the pathophysiology of NDDs [29]. Associations between GI dysfunction and NDDs might be explained by changes in stress-related and gut–brain biomarkers. Children with NDDs are reported to present alterations in stress system activity and in the composition of the gut microbiota [30]. Furthermore, FGIDs in children with NDDs have been associated with symptoms of stress and anxiety, internalizing and externalizing problems, sensory hyperresponsivity stress and other contributing factors [31]. 

Internalizing (INT) and externalizing (EXT) problems are a common comorbidity in children with NDDs. Internalizing problems include anxiety, sadness, social withdrawal, and fearfulness, while externalizing symptoms comprise ADHD symptomatology, oppositional defiant disorder (ODD), conduct disorder (CD), substance use, noncompliance and aggression [32,33,34]. Elevated levels of INT and EXT symptoms are closely linked to deficits in social and academic abilities, the development of psychiatric comorbidities in adulthood and lower quality-of-life outcomes [35,36,37]. 

The stress system consists of the hypothalamic–pituitary–adrenal (HPA) axis and the sympathetic–autonomic nervous system (ANS). It is a regulation system with a main function of responding to environmental and psychological stressors. The HPA axis and the ANS interact with other parts of the central nervous system (CNS) and organs-target, producing glucocorticoids and catecholamines, as an adaptive response to stressful circumstances [38]. Hair and salivary cortisol are considered reliable biomarkers of the HPA axis activity [39]. Hyper- or hypo-activation of the HPA axis is associated with homeostatic alterations and may be related with a variety of clinical somatic and mental health manifestations [40]. Leptin is a peptide hormone released from the adipose tissue, involved in the function of the HPA axis. Studies show that leptin modulates the secretion of corticotropin-releasing hormone, inhibits the secretion of steroids and blunts the HPA axis activation [41].

Environmental and psychosocial stressors have an impact on the function of the GI tract due to alterations in the stress system and the gut–brain axis [42,43]. Additionally, a recent study indicates that psychological distress can impact both systemic and gut immunity, which is increasingly acknowledged as a pathophysiological characteristic of FGID [44]. There are published data regarding the deviation of the HPA axis in children with NDDs [6,45]. The dysregulation of stress-related mechanisms may play a significant role in NDDs pathophysiology [46,47,48,49,50,51,52].

There are limited studies examining the association of FGIDs, NDDs, stress-related biomarkers and externalizing and internalizing behavioral symptoms in school-aged children and adolescents. The aim of the current study was to assess the presence of FGIDs in children and adolescents with ASD and ADHD in comparison to a sample of typically developed children. Furthermore, to investigate the potential role of co-morbid internalizing and externalizing behavioral problems, as well as the role of stress-related biomarkers in children with FGIDs and ASD/ADHD, we measured salivary and hair cortisol concentrations and plasma leptin concentrations in children with ASD and ADHD and children of typical development (TD). We hypothesized that children with ADHD and ASD would indicate a higher prevalence of FGIDs compared to TD children and that internalizing/externalizing problems and stress-related biomarkers would connect these comorbidities.

## 2. Materials and Methods

### 2.1. Participants

The current study was a case-control study conducted between September 2017 and July 2023. A total number of 120 Greek preschool and school-aged children of both sexes, between 4 and 12 years old (34.1% girls; 65.9% boys, mean age: TD 8.07 years (SD 2.03), ASD 8.56 years (SD 2.03), ADHD 7.55 years (SD 1.85)), participated in the study. Two clinical groups (40 children with ASD diagnosis, 40 children with ADHD diagnosis) and a typical development group (40 TD children) were enrolled in the study. All children were of normal intelligence. The descriptive characteristics are presented in Table 1.

The clinical groups were recruited from scheduled appointments in the Unit of Developmental and Behavioral Pediatrics of the First Department of Pediatrics, “Aghia Sophia” Children’s Hospital, School of Medicine, National and Kapodistrian University of Athens, Greece. The TD group was recruited from regular outpatient pediatric clinics and were screened for neurodevelopmental and mental health symptoms. The study included all children who participated with their parents’ written informed consent. All procedures adhered to the principles of the Declaration of Helsinki and were approved by both the Scientific and Ethics Committees of the ‘Aghia Sophia’ children’s hospital (Institutions Review Board: 17383/26 September 2017)

#### 2.1.1. Clinical Diagnoses

Throughout the procedure, a comprehensive medical and developmental history was acquired via a clinical interview conducted with the parents/caregivers. Additionally, a thorough clinical examination and developmental assessment were conducted on all children, including those in the TD group, by a board-certified pediatrician. Clinical diagnoses were determined by a Developmental–Behavioral Pediatrician with an extensive clinical and research background, alongside a multidisciplinary team of clinical experts, in accordance with standardized criteria based on the DSM-5 [2]. The diagnosis of FGIDs was based on the Rome criteria III-Greek version [26], a symptom-based diagnostic classification established by expert consensus.

#### 2.1.2. Inclusion and Exclusion Criteria

The children included in the study met the DSM-5 criteria and were of average intelligence. The body mass index (BMI) of all participants was from average to overweight. The group of children diagnosed with ADHD consisted solely of individuals with a diagnosis of ADHD-combined presentation (inattention and hyperactivity). The study employed specific exclusion criteria to ensure the homogeneity of the participant sample and to minimize potential confounding factors. These criteria included (1) comorbidity of ASD and ADHD, (2) underlying chronic neurological, chronic systemic, genetic or chromosomal disorder, (3) severe intellectual or psychiatric disorders in the parents and (4) inability to speak the Greek language. Individuals meeting any of these criteria were excluded from the study.

#### 2.1.3. Tools of Assessment

All children underwent comprehensive interviews conducted by a developmental–behavioral pediatrician to assess symptomatology for clinical diagnoses based on DSM-5 criteria [2]. Additionally, the Athina test was administered to evaluate learning and cognitive abilities [53]. All parents completed self-reported questionnaires for their children’s behavior, ADHD symptomatology and functional gastrointestinal disorders.

##### Child Behavior Checklist/6–18 (CBCL) for Parents

The Child Behavior Checklist/6–18 for parents were administered to all groups for screening behavioral and emotional comorbid conditions [32,54]. The Child Behavior Checklist, now referred to as the Achenbach System of Empirically Based Assessment, serves as a parent-reported form designed to assess emotional, behavioral and social problems in children. Structured around questions associated with problems across eight different categories, namely, anxious/depressed, withdrawn/depressed, somatic complaints (internalizing problems), social problems, thought problems, attention problems, rule-breaking behavior and aggressive behavior (externalizing problems), the CBCL also includes scales indicating scores related to disorders outlined in the Diagnostic and Statistical Manual of Mental Disorders (DSM-5), such as anxiety, oppositional defiant disorder, conduct problems, somatic problems, affective problems and attention deficit disorder [15]. Numerous studies have demonstrated the CBCL’s high reliability in correlation with psychological diagnoses [55].

##### The Questionnaire on Pediatric Gastrointestinal Symptoms—Greek Version of Rome III

The Roma Foundation working team has assessed a specific tool for the clinical identification of FGIDs in the pediatric population, the Questionnaire on Pediatric Gastrointestinal Symptoms-Rome III (QPGS-RIII), which includes and categorizes 10 gastrointestinal disorders [56,57]. All questionnaires were completed by parents regardless of the participant’s age. Utilizing a series of Likert-type scales comprising 71 questions, the questionnaire evaluates the frequency, severity and duration of symptoms. Furthermore, it can be scored to determine if a patient meets the criteria for specific functional gastrointestinal disorders. Following the completion of the questionnaire, a coding system is employed to derive provisional diagnoses based on the responses provided [58].

##### Athina Test

The Athina Test is a clinical tool administered individually to assess learning difficulties [53]. It offers standardized measurements across various abilities that reflect different aspects of learning challenges. Utilizing psychometric scales, the test evaluates the developmental level and pace of the child across five key areas: cognitive ability (including linguistic analogies and vocabulary), sequential direct memory (such as numerical memory), the completion of incomplete expressions (like sentence completion), grapho-phonemic awareness (involving graph distinction and phoneme composition) and neuro-psychological maturity (including right–left perception). The subscales of the cognitive ability scale were used as an initial evaluation of children’s mental capacity [59].

## 3. Biological Markers

### 3.1. Salivary Samples Collection and Analysis

The study assessed various neuroendocrine parameters associated with stress system activity in all participants: (a) Diurnal variation in salivary cortisol: Saliva samples were collected at home, supervised by caregivers on a regular weekend, at six different time points, upon awakening at approximately 8:00 a.m. (C1), 30 min after waking up (C2), at 12:00 p.m. (C3), at 3:00 p.m. (C4), at 6:00 p.m. (C5) and at 9:00 p.m. (C6). (b) Cortisol Awakening Response (CAR): This was indicated by the increase in salivary cortisol from waking to 30 min later (C1–C2). (c) The slope for cortisol, which was estimated with the following type: Slope¼C3–C2. (d) The salivary cortisol and area under the curve with respect to the ground (AUCg) and with respect to an increase (AUCi) as measures of total cortisol. AUCg is indicative of the total hormonal output, while AUCi is related to the sensitivity of the stress system [60].

Salivary samples were collected using the Sarstedt Salivette^®^ system (Sarstedt Inc., Newton, NC, USA). Parents and children received face-to-face demonstrations and detailed written and oral instructions on the saliva collection procedure. Parents were advised to collect samples after the child rinsed their mouth with water, ensuring a minimum of 30 min had elapsed since eating, drinking, tooth-brushing, intense play or exercise. Parents were also asked to record the start time of sample collection to ensure accuracy. Children were instructed to keep the cotton swabs in their mouths for at least 2 min to ensure saturation. Subsequently, the swabs were placed in labeled plastic tubes and stored in the refrigerator at 0–4 °C. The samples were returned to the researcher within 2 days for further processing. Saliva was extracted from the cotton swab by centrifugation at 3500× *g* for 5 min, and aliquots were stored at −85 °C until analysis. Aliquots from each saliva sample were utilized for cortisol assessment, with all analyses conducted in duplicate. Salivary cortisol concentrations were determined using a chemiluminescence immunoassay (Elecsys Cortisol, Roche Diagnostics International AG, Rotkreuz, Switzerland), following the manufacturer’s instructions. The analytical sensitivity was 0.054 μg/dL, and both intra- and inter-assay coefficients of variation (CV) were 3.0% and 11.8%, respectively. Salivary cortisol levels were expressed in nmol/L.

### 3.2. Hair Cortisol Samples Collection and Analysis

The researcher cut a hair strand of 1–2 cm in length from the posterior vertex area of the participants (weighing approximately 10 mg per cm), as close as possible to the scalp, using scissors. The hair was enclosed in sealed plastic packages marked with an identification number and were stored at room temperature until analysis. All analyses were performed in the Laboratory of Endocrinology of “Aghia Sophia” Children’s Hospital according to the hair cortisol procedure. The hair sample was weighed (minimum of 25 and maximum of 30 mg of hair/sample) and placed in a homogenization tube (Precellys Lysing tubes, Bertin Instruments, Montigny-le-Bretonneux, France) along with corresponding beads (two large and seven small beads for each sample). The test tubes were then placed in a homogenizer Minilys (Bertin Instruments, Montigny-le-Bretonneux, France) to lyse the hair. Each sample was centrifuged at least seven times (60 s in 5000 rpm). When the powder-form was ready, 1 mL of methanol was added in each sample, and all tubes were placed in a shaker for 16 h at room temperature. The following day, the tubes were centrifuged at 13,000 rpm for 10 min, and the supernatant was transferred in glass tubes in order to fully evaporate the methanol. Finally, after approximately four days, the samples were diluted in 100 μL phosphate-buffered saline (pH 8.0, 1× PBS) and were then vigorously vortexed for 1 ½ min until they were well mixed for the analysis. Eventually, the samples were measured by using an automated electrochemiluminescence immunoassay via an automated analyzer Cobas e411—Roche Diagnostics. The hair cortisol levels were expressed in pg/mg [61].

Hair cortisol concentrations reflect exposure to chronic stress. Hair grows approximately 1 cm per month [62].

### 3.3. Leptin Samples Collection and Analysis

Peripheral blood samples were drawn and centrifuged at 2300 rpm, at room temperature, for 10 min. The resulting plasma was frozen at −80 °C. Concentrations of the dissoluble receptor of human plasma were measured using a commercial Sandwich ELISA (BioVendor-Laboratorni medicina a.s., Brno, Czech Republic) and an HRP-labelled antibody method by a Biovendor kit (lower limit of detection 0.2 ng/mL, intra-assay CV = 5.9%, inter-assay CV = 5.6%). The samples were run in concordance with the instructions of the kit protocol. All children’s BMIs were calculated by a body mass index specific for age and sex (BMI, calculated in kg/m^2^) [63,64].

## 4. Statistical Analysis

The data are presented as the frequencies (%) for categorical variables and as the Mean (Standard Deviation, SD) or Median (Interquartile Range, IQR), according to the distribution, for the quantitative variables. The normality of the distribution was assessed using the Kolmogorov–Smirnov test. A chi square test was used to assess differences between categorical variables. One-Way ANOVA or Kruskal Wallis tests were used to evaluate significant differences between quantitative variables and the three study groups (control, ADHD, ASD). Finally, an Independent Samples T-Test or Mann Whitney U test was used to evaluate significant differences between quantitative variables and two different groups (existence of gastrointestinal problems vs. nonexistence of gastrointestinal problems), either for the whole sample or separately for the three study groups (control, ADHD, ASD). The statistical package SPSS version 26 was used, and the level of significance was 0.05.

## 5. Results

Table 1 shows descriptive characteristics and measurements separately for the three study groups (control, ADHD, ASD). Significant differences were observed for sex; there were more boys than in the control group (*p* < 0.0001) and, as expected, in the ASD group. Relevant to the Achenbach questionnaire, the values were significantly lower for the control group compared to the ADHD and ASD groups, either for the total score or for the internalizing and externalizing scores (*p* < 0.0001, *p* < 0.0001, *p* = 0.005, respectively). Participants who had INT and EXT behavioral clinical scores were less common in the control group compared to the ADHD and ASD groups, either for the total score or for the internalizing and externalizing scores (*p* = 0.003, *p* = 0.020, *p* = 0.037, respectively). According to the gastrointestinal problems, in the ASD group, their existence was greater than that of the control group (*p* = 0.001). The mean value of the AUCg was significantly lower for the ADHD group compared to the control and ASD groups (*p* < 0.0001), while the value of hair cortisol was significant higher for the control group compared to that for the ADHD and ASD groups (*p* < 0.0001). In this study, no significant differences were found between preadolescent and adolescent children with FGIDs, as well as between males and females with FGIDs, possibly due to the small sample.

Table 2 presents the proportion of the existence or nonexistence of gastrointestinal problems separately for the three study groups (control, ADHD, ASD). Generally, aerophagia was more frequently observed for all the groups. For the control group, the second most frequent gastrointestinal problem was functional constipation, for the ADHD group, it was functional abdominal pain syndrome and for the ASD group, it was non-retentive fecal incontinence.

Table 3 shows values for cortisol (AUCg, AUCi, CAR, morning cortisol, evening cortisol, hair cortisol), leptin and the Achenbach questionnaire, separately for children with and without gastrointestinal problems. Significant differences were observed only for the total, the internalizing score and the externalizing score, where children with gastrointestinal problems scored higher in all three scales compared to children without gastrointestinal problems (*p* < 0.0001, *p* < 0.0001, *p* = 0.041, respectively).

Table 4 presents values for cortisol (AUCg, AUCi, CAR, morning cortisol, evening cortisol, hair cortisol), leptin and the Achenbach questionnaire, separately for children with and without gastrointestinal problems and for the three study groups (control, ADHD, ASD). In the ADHD group, children with gastrointestinal problems scored higher both in the internalizing scale and the total scale than those without problems (*p* < 0.0001 and *p* = 0.013, respectively). Children with gastrointestinal problems in the control group had significantly lower values of AUCg in contrast to those without problems (*p* = 0.039).

## 6. Discussion

The present study aimed to contribute to the field of FGIDs in NDDs and, more specifically, to investigate the presence of functional gastrointestinal disorders in a clinical sample of children with Autism and ADHD, compared to healthy volunteers, and to explore possible differences in behavioral and neuroendocrine measures between those with and without such comorbidities.

### 6.1. Internalizing and Externalizing Problems

The study showed that the ADHD and ASD groups had significantly higher clinical scores of internalizing and externalizing problems compared to the TD group, which was almost expected. A study of Pandolfi et al. found that children with ASD showed anxiety and depression combined with increased total problems [65]. Another study demonstrated higher total problem scores in school-aged children with ASD and in children with lower cognitive abilities [66]. Biederman and his colleagues found that 44% of children with ADHD had elevated rates of comorbid psychopathology [67]. In children with NDDs, internalizing problems may be associated with HPA axis hyperactivation, while externalizing problems correlate with HPA axis hypoactivation [68]. Sanjaya et al. have found that cortisol levels are elevated in individuals who are at risk of internalizing, externalizing and attention disorders [69]. In addition, children with gastrointestinal problems scored higher values on the total, internalizing and externalizing scales. Ferguson et al. showed that older children with autism and internalizing/externalizing problems were more likely to suffer from constipation and abdominal discomfort, while younger children usually present nausea [70].

Overall, our study revealed that the ADHD group with comorbid gastrointestinal problems scored higher both in the internalizing and total scales. This result is in accordance with the literature suggesting that children with ADHD and internalizing problems exhibit elevated daily hassles including emotional distress, anxiousness, social difficulties, low self-esteem and somatic turbulences such as GI disorders, stomachaches, migraine and sleep disorders [71,72]. Melegari et al. have found that internalizing problems are capable of exacerbating the ADHD symptomatology, especially when anxiety exists as a comorbidity and is associated with attention deficit difficulties [73].

### 6.2. GI Measurements

It has been estimated that 9–91% of children with ASD report gastrointestinal problems [74]. According to the statistical analysis in this study, the most frequent gastrointestinal symptom is aerophagia (in both clinical and TD groups). Moreover, in the control group, the second most frequent gastrointestinal problem is functional constipation. Regarding the ADHD group, the second most frequent GI symptom is functional abdominal pain syndrome, and for the ASD group, it is non-retentive fecal incontinence. A recent metanalysis of Lasheras I. et al. reports that the most frequent gastrointestinal symptoms among children with ASD are constipation (87.5%), and the second most frequent GI symptom is aerophagia (75%), alongside abdominal pain (75%) and vomiting (75%) [27]. There is an association between the severity of ASD and the presence of GI symptoms [75]. However, in a Danish study, there was no significant difference in GI disorders between a group with atypical autism and a TD group [76]. In the study of Maenner et al., there was no association between the presence of GI problems and stereotypical or repetitive behaviors [30].

### 6.3. Salivary and Hair Cortisol Measurements

Our findings documented no differences in the daily secretion of cortisol between the clinical groups and the TD children; nevertheless, the mean value of the AUCg was significant lower for the ADHD group compared to the control and ASD groups. Even though ADHD has been associated with attenuated HPA axis activity, Angeli et al. have found no differences in CAR and slope cortisol levels between the three ADHD subtypes [77]. Moreover, Anesiadou et al. have made no significant findings regarding the salivary cortisol diurnal secretion in groups of ASD and ADHD children compared to TD children [6]. Studies have shown that participation in clinical procedures is considered as a perceived stressor factor and may lead to an increase in cortisol during the experimental part of the study [78,79,80]. Similarly, the ADHD and ASD groups showed lower hair cortisol values compared to the TD group. There are plenty of factors that contribute to the age-related cortisol response to psychosocial stress, such as sociocognitive maturation. It is known that children with ASD and ADHD have deficits in social and emotional responses. Non-adaptive social responses, the lack of a theory of mind and a cognitive capacity, characterized by profound difficulties in social interaction and communication, prevent an appropriate response to stressful stimuli and environmental demands [81,82].

Activation of the HPA axis coordinates adaptation to stress, and cortisol is considered essential for normal neurodevelopment. The low levels of cortisol in children with ADHD are possibly correlated with hypoarousal of the HPA axis or indicate a delay in daily response secretion [83]. In the study by Pauli-Pott et al., it is highlighted that low hair cortisol in preschool children is a predictor of ADHD developmental disorder and comorbidities with internalizing behavioral problems at school age. [84]. In Ogawa S. et al.’s study, a positive correlation between cognitive functions and hair cortisol was present only in TD children, and a negative correlation was present in the ASD group. These results underly the relation between elevated chronic stress hormones in children with ASD and cognitive dysfunctions [85]. However, other studies suggest that the coordination between the HPA axis and the sympathetic nervous system in generating the physiological adaption of the stress response remains a topic to debate [38].

Moreover, children with GI problems in the control group had significantly lower values in AUCg in contrast to those without GI problems. The presence of GI disorders can be affected by the autonomic nervous system by it regulating the release of stress hormones [86]. Studies have also demonstrated that stress factors can impair the gut’s permeability and mobility and lead to constipation or diarrhea due to the deviation of the enteric nervous system [87]. Finally, alterations in the GI tract provoked by stress factors have been found to corelate with a higher sensibility of individuals with GI disorders and comorbidities with brain health issues [88].

### 6.4. Leptin Measurements

Serum plasma leptin measurements revealed no significant differences in plasma leptin values when the clinical group was compared to the TD group, both with or without GI disorders. Flores-Dorantes M.T. et al. found in their study that leptin and its receptors may have a possible role in neurodevelopment by their effect in neuroinflammation and neurodegeneration [89]. Some cross-sectional studies revealed confusing data while investigating associations between leptin and NDDs [90,91]. On the other hand, Prosperi et al. have found no correlation between the levels of leptin and developmental regression in children with ASD [92]. Finally, some studies demonstrate that plasma leptin levels are higher in children with ASD and normal or above-average BMIs [41].

## 7. Strengths and Limitations

The strength points of our study are summed up as follows: To the best of our knowledge, it is the first time that NDDs, INT and EXT behavioral problems, FGDIs and stress-related biomarkers have been measured in a school-aged population. The clinical groups were not under pharmacological treatment, so the differences in our measurements may be a result of NDDs comorbidities. The present study contributes to existing evidence regarding the activity of the HPA axis in children with NDDs.

However, there are some limitations: The study sample is relatively small, partly due to the difficulties in obtaining samples from young children with developmental and behavioral disorders. This was a cross-sectional study; thus, no etiological relations can be established. The recruited sample was not equal among the groups. The number of the boys is higher than that of the girls, but this is in accordance with the international literature, which reveals that NDDs are more common between males. The assessment of GI relied upon parent-report questionnaires (Greek Version Rome III), and GI symptoms were not clinically confirmed by a pediatric gastroenterologist.

## 8. Conclusions

In the current study, children with Autism and ADHD showed an increased prevalence of FGIDs and internalizing/externalizing problems. Aerophagia was the most frequent gastrointestinal symptom in all three groups, while the second most frequent in the TD group was functional constipation, functional abdominal pain syndrome for the ASD group and non-retentive fecal incontinence for the ADHD group, respectively. It is remarkable that children with ADHD and co-morbid symptoms of FGIDs had higher scores on the internalizing and total scales compared to their peers without FGIDs. Stress-related biomarkers present differences between the clinical groups: The ADHD group had significantly lower AUCg values compared to the TD and ASD groups. The TD group with FGIDs problems showed a lower cortisol secretion of AUCg compared to those without FGIDs. Finally, plasma serum leptin analysis revealed no differences among all groups. Longitudinal studies with a greater number of children with NDDs and FGIDs may elucidate the behavioral and biological pathways linking this common comorbidity. Nevertheless, in our study, no significant differences have been found in salivary cortisol or serum leptin concentrations between children with NDDS with or without FGIDs. Further prospective and longitudinal studies, focused on stress biomarkers, may reveal the pathophysiological mechanisms underlying these relations.

## Figures and Tables

**Table 1 nutrients-16-01538-t001:** Descriptive Characteristics and separate measurements for the three study groups.

	Control	ADHD	ASD	*p*-Value
**Sex N (%)** − Boys − Girls	20 (50%) * 20 (50%) *	23 (57.5%) 17 (42.5%)	38 (90%) * 4 (10%) *	<0.0001
**Age Median (IQR)** **Mean (SD)**	8.1 (3.53) 8.07 (2.03)	7.2 (1.58) 7.55 (1.85)	8.45 (3.28) 8.56 (2.03)	0.069
**Existence of gastrointestinal problems N (%)** − No − Yes	31 (79.5%) * 8 (20.5%) *	25 (64.1%) 14 (35.9%)	15 (39.5%) * 23 (60.5%) *	0.001
**BMI z-score Median (IQR)**	−0.18 (0.34)	−0.12 (1.52)	−0.19 (0.18)	0.149
**AUCg Mean (SD)**	1.73 (0.68) *	1.05 (0.94) *^	1.44 (0.55) ^	<0.0001
**AUCi Median (IQR)**	−2.11 (3.00) *^	−0.33 (3.11) *	−2.03 (3.08) ^	0.066
**Cortisol Hair Median (IQR)**	7.84 (7.56) *^	3.99 (2.16) *	4.45 (2.36) ^	<0.0001
**Leptin Median (IQR)** − Boys − Girls	2.70 (14.68) 6.72 (4.45)	2.85 (3.65) 4.94 (9.24)	2.21 (7.47) 13.58 (20.58)	0.663 0.491
**Internalizing Median (IQR)**	54 (62.5) *	79 (45)	94 (16) *	<0.0001
**Externalizing Median (IQR)**	63.5 (60) *	87 (41.5) *	76 (19.75)	0.005
**Total Achenbach score Median (IQR)**	50 (64.25) *^	89 (28.5) *	89 (21.25) ^	<0.0001
**Internalizing Categories N (%)** − Normal − Borderline − Clinical	20 (62.5%) * 5 (15.6%) 7 (21.9%) *	18 (54.5%) 3 (9.1%) 12 (36.4%)	5 (20.8%) * 5 (20.8%) 14 (58.3%) *	0.020
**Externalizing Categories N (%)** − Normal − Borderline − Clinical	24 (75%) * 5 (15.6%) 3 (9.4%) *	14 (42.4%) * 5 (15.2%) 14 (42.4%) *	12 (50%) 5 (20.8%) 7 (29.2%)	0.037
**Total Categories N (%)** − Normal − Borderline − Clinical	22 (68.8%) * 5 (15.6%) 5 (15.6%) *	15 (45.5%) 3 (9.1%) * 15 (45.5%) *	6 (25%) * 9 (37.5%) * 9 (37.5%)	0.003

*, ^ shows the significant differences between the groups. One-Way ANOVA, Kruskal Wallis, *p* < 0.05. ADHD = Attention Deficit Hyperactivity Disorder, ASD Autism Spectrum Disorder. SD = Standard Deviation, IQR = Interquartile Range.

**Table 2 nutrients-16-01538-t002:** Proportion of the existence of gastrointestinal problems separately for the three study groups.

	Control N (%)	ADHD N (%)	ASD N (%)
**Functional Dyspepsia** − No − Yes	38 (97.4) 1 (2.6)	38 (97.4) 1 (2.6)	36 (97.3) 1 (2.7)
**Irritable Bowel Syndrome** − No − Yes	37 (94.9) 2 (5.1)	38 (97.4) 1 (2.6)	36 (97.3) 1 (2.7)
**Abdominal Migraine** − No − Yes	37 (94.9) 2 (5.1)	36 (92.3) 3 (7.7)	36 (97.3) 1 (2.7)
**Functional Abdominal Pain** − No − Yes	38 (97.4) 1 (2.6)	38 (97.4) 1 (2.6)	36 (97.3) 1 (2.7)
**Functional Abdominal Pain Syndrome** − No − Yes	39 (100) 0 (0)	35 (89.7) 4 (10.3)	35 (94.6) 2 (5.4)
**Functional Constipation** − No − Yes	36 (92.3) 3 (7.7)	37 (94.9) 2 (5.1)	31 (83.8) 6 (16.2)
**Non-Retentive Fecal lncontinence** − No − Yes	39 (100) 0 (0)	39 (100) 0 (0)	30 (81.1) 7 (18.9)
**Aerophagia** − No − Yes	33 (84.6) 6 (15.4)	31 (79.5) 8 (20.5)	26 (70.3) 11 (29.7)
**Cyclic Vomiting Syndrome** − No − Yes	39 (100) 0 (0)	39 (100) 0 (0)	37 (100) 0 (0)
**Adolescent Rumination Syndrome** − No − Yes	39 (100) 0 (0)	38 (97.4) 1 (2.6)	37 (100) 0 (0)

**Table 3 nutrients-16-01538-t003:** Measurements for groups (N = 120) with or without gastrointestinal problems.

	Non-Existence of Gastrointestinal Problems	Existence of Gastrointestinal Problems	*p*-Value
**AUCg Mean (SD)**	1.46 (0.72)	1.28 (0.47)	0.180
**AUCi Median (IQR)**	−1.61 (3.79)	−1.45 (2.44)	0.878
**Cortisol Hair Median (IQR)**	5.46 (3.81)	4.53 (3.14)	0.132
**CAR Median (IQR)**	−0.05 (0.25)	−0.06 (0.15)	0.882
**Morning Cortisol Median (IQR)**	0.23 (0.04)	0.24 (0.24)	0.589
**Evening Cortisol Median (IQR)**	0.05 (0)	0.05 (0)	0.366
**Leptin Median (IQR)**	3.21 (6.60)	3.53 (10.17)	0.833
**Internalizing Median (IQR)**	58 (55.25)	95 (11)	<0.0001
**Externalizing Median (IQR)**	67 (50.75)	89 (44)	0.041
**Total Achenbach Score Median (IQR)**	73 (58)	89 (17)	<0.0001

Independent Samples T-Test, Mann Whitney, *p* < 0.05. SD = Standard Deviation, IQR = Interquartile Range.

**Table 4 nutrients-16-01538-t004:** Measurements for groups with or without gastrointestinal problems separately for the three study groups.

	Control		*p*-Value	ADHD		*p*-Value	ASD		*p*-Value
	Non-Existence of Gastrointestinal Problems	Existence of Gastrointestinal Problems		Non-Existence of Gastrointestinal Problems	Existence of Gastrointestinal Problems		Non-Existence of Gastrointestinal Problems	Existence of Gastrointestinal Problems	
**AUCg** **Mean (SD)**	1.84 (0.74)	1.42 (0.27)	0.039	1.05 (0.65)	1.02 (0.61)	0.876	1.49 (0.64)	1.38 (0.51)	0.554
**AUCi** **Median (IQR)**	−1.90 (3.30)	−2.31 (2.92)	0.813	0.0 (3.98)	−0.76 (2.46)	0.400	−2.00 (3.81)	−2.06 (1.97)	0.906
**Cortisol Hair Median (IQR)**	7.94 (7.04)	7.56 (7.08)	0.695	4.47 (2.81)	3.64 (1.48)	0.191	4.37 (2.00)	4.76 (3.27)	0.699
**CAR Median (IQR)**	−0.10 (0.49)	−0.75 (0.56)	0.942	0 (0.29)	−0.03 (0.11)	0.672	−0.15 (0.35)	−0.07 (0.15)	0.300
**Morning Cortisol Median (IQR)**	0.26 (0.28)	0.27 (0.23)	0.478	0.12 (0.48)	0.13 (0.027)	0.942	0.24 (0.38)	0.26 (0.24)	0.836
**Evening Cortisol Median (IQR)**	0.05 (0)	0.05 (0)	0.575	0.05 (0)	0.05 (0)	0.806	0.05 (0)	0.05 (0)	0.680
**Leptin** **Median (IQR)**	3.84 (10.75)	4.29 (2.69)	0.976	3.16 (5.18)	3.64 (7.04)	0.817	2.35 (7.05)	2.70 (18.84)	0.616
**Internalizing Median (IQR)**	44 (57.5)	84 (50)	0.076	65.5 (38.25)	95 (18.25)	<0.0001	90 (30.5)	96.5 (8)	0.172
**Externalizing Median (IQR)**	65 (60)	38 (74)	0.800	83 (43)	92 (27.25)	0.077	81.5 (45)	81.5 (45.5)	0.752
**Total Achenbach Score** **Median (IQR)**	38 (66)	87 (51)	0.216	77 (43.75)	98 (18.5)	0.013	84 (49)	89 (10.25)	0.403

Independent Samples T-Test, Mann Whitney, *p* < 0.05. ADHD = Attention Deficit Hyperactivity Disorder, ASD = Autism Spectrum Disorder, SD = Standard Deviation, IQR = Interquartile Range.

## Data Availability

The data that support the findings of this study are available from the corresponding author, Panagiota Pervanidou, upon reasonable request. The data are not publicly available due to their containing information, which could compromise the privacy of research participants.

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
