# Peer review of "Functional Gastrointestinal Symptoms in Children with Autism and ADHD: Profiles of Hair and Salivary Cortisol, Serum Leptin Concentrations and Externalizing/Internalizing Problems"

_nutrients, 2024, doi:10.3390/nu16101538_

Round 1
Reviewer 1 Report
Comments and Suggestions for Authors
This is an interesting reaserch article with adequate novelty and quality. However, some points should be addressed.
- The numbering order of authors affiliations should be revised.
- The authors should add subheading in the abstract based on the guidelines of the journal.
- At the end of the Abstract, 1-2 sentences with the main conclusions of the study and the potential future perspectives based on its results should be added.
- The sentence in lines 61-62 "Several studies have shown that FGIDs 61 may have an increased prevalence in children with NDDs" needs a bit more analysis in this point.
- Relevant references should be added concerning the two sentences in lines 81-84 "Hair and salivary cortisol are considered reliable biomarkers of the HPA axis activity. Hyper or hypo-activitation of the HPA axis is associated with homeostatic alterations and may be related with a variety of clinical somatic and mental health manifestations."
- Relevant references should be added concerning the sentence in lines 83-84 "There are published data regarding the deviation of the HPA axis in children with NDDs."
- Before reporting the aim of the study at the end of the Introduction section, the authors should emphasize the scientific literature gap that exists and which will be covered by the present study.
- In lines 110-111, the mean values of age should be accompanied by standard deviation values.
- Relevant references should be added concerning the sentence in lines 147-149 "All children underwent comprehensive interviews conducted by a developmental behavioral pediatrician to assess symptomatology for clinical diagnoses based on DSM-V criteria."
- Again, relevant references should be added concerning the sentence in lines 149-150 "Additionally, the Athina test was administered to evaluate learning and cognitive abilities.".
- The symbol % should be added concerning the percentage values included in Table 1.
- At the beginning of the Discussion section the authors should emphasize the scientific literature gap that exists and which has been covered by the present study.
- The 1st paragraph in section 4.3 is a bit long and it should be split into two smaller paragraphs.
- Some more spesific suggestions should be proposed at the end of the Conclusion section concerning the future studies that could be performed in the topic of the present study.
- There are several typos and grammar/syntax errors throughout the manuscript which should be revised.
Comments on the Quality of English LanguageThere are several typos and grammar/syntax errors throughout the manuscript which should be revised. English language editing is highily recommended.
Author Response
Response to Reviewer 1 Comments
We appreciate very much the Reviewers’ interest in our work, for the helpful comments as well as the effort and time put into the review of this manuscript. All comments have been carefully considered and responded point by point.
Comment 1: The numbering order of authors affiliations should be revised.
Response 1.1: We thank the Reviewer for this comment. We have revised the authors affiliations according to the guidelines.
Comment 2: The authors should add subheading in the abstract based on the guidelines of the journal.
Response 1.2: We thank the Reviewer for this comment. We have revised the subheading in the abstract according to the journal’s guidelines.
Comment 3: At the end of the Abstract, 1-2 sentences with the main conclusions of the study and the potential future perspectives based on its results should be added.
Response 3: We thank the Reviewer for this comment. Due to limited word count in the abstract, we only added one sentence...”Future prospective studies including a greater number of children may elucidate the biological pathways linking these comorbidities “ (lines 41-43).
Comment 4: The sentence in lines 61-62 "Several studies have shown that FGIDs 61 may have an increased prevalence in children with NDDs" needs a bit more analysis in this point.
Response 4: We thank the Reviewer for this comment. We added a revised text with references as following: Several studies have shown that FGIDs may have an increased prevalence in children with NDDs in comparison with the general population. According to Lasheras et al., the prevalence of GI disorders in children with ASD is 13%–57% while other studies ranged them between 9% and 91% [27]. The strong correlation between GI dysfunction and NDDs could be explained through the microbiota–gut–brain axis (MGBA) [28]. The MGBA has been coined to describe a bidirectional communication pathway between the microbiota, gut and central nervous system (CNS) which can explain the pathophysiology of NDDs [29]. (lines 71-78)
Comment 5: Relevant references should be added concerning the two sentences in lines 81-84 "Hair and salivary cortisol are considered reliable biomarkers of the HPA axis activity. Hyper or hypo-activitation of the HPA axis is associated with homeostatic alterations and may be related with a variety of clinical somatic and mental health manifestations."
Response 5: We thank the Reviewer for this comment. We added relevant references as following: Hair and salivary cortisol are considered reliable biomarkers of the HPA axis activity [39]. (line 98) Hyper or hypo-activitation of the HPA axis is associated with homeostatic alterations and may be related with a variety of clinical somatic and mental health manifestations [40]. (line 100)
Comment 6: Relevant references should be added concerning the sentence in lines 83-84 "There are published data regarding the deviation of the HPA axis in children with NDDs."
Response 6: We thank the Reviewer for this comment. We added relevant references as following: There are published data regarding the deviation of the HPA axis in children with NDDs [45,6]. (lines 108-109)
Comment 7: Before reporting the aim of the study at the end of the Introduction section, the authors should emphasize the scientific literature gap that exists and which will be covered by the present study.
Response 7: We thank the Reviewer for this comment. We added the revised text as following: There are limited studies examining the association of FGIDs, NDDs, stress related biomarkers and externalizing and internalizing behavioral symptoms in school aged children and adolescents (lines 111-113).
Comment 8: In lines 110-111, the mean values of age should be accompanied by standard deviation values.
Response 8: We thank the Reviewer for this comment. We added the standard deviation values as following: TD (2.03), ADHD (1.85), ASD (2.03) (lines 127-128)
Comment 9: Relevant references should be added concerning the sentence in lines 147-149 "All children underwent comprehensive interviews conducted by a developmental behavioral pediatrician to assess symptomatology for clinical diagnoses based on DSM-V criteria."
Response 9: We thank the Reviewer for this comment. We added relevant references as following: All children underwent comprehensive interviews conducted by a developmental -behavioral pediatrician to assess symptomatology for clinical diagnoses based on DSM-5 criteria [2]. (line 165)
Comment 10: Relevant references should be added concerning the sentence in lines 149-150 "Additionally, the Athina test was administered to evaluate learning and cognitive abilities."
Response 10: We thank the Reviewer for this comment. We added relevant references as following: Additionally, the Athina test was administered to evaluate learning and cognitive abilities [53] (line 166)
Comment 11: The symbol % should be added concerning the percentage values included in Table 1.
Response 11: We thank the Reviewer for this comment. We added the symbol % in Table 1. (line 297)
Comment 12: At the beginning of the Discussion section the authors should emphasize the scientific literature gap that exists and which has been covered by the present study.
Response 12: We thank the Reviewer for this comment. The revised text is the following: The present study aimed to cover the scientific gap in the field of FGIDs in NDDs and more specific to investigate the presence of functional gastrointestinal disorders in a clinical sample of children with Autism and ADHD, compared to healthy volunteers, and to explore possible differences in behavioral and neuroendocrine measures between those with and without such comorbidities. (line 330-334)
Comment 13: The 1st paragraph in section 4.3 is a bit long and it should be split into two smaller paragraphs.
Response 13: We thank the Reviewer for this comment. We split the 1st paragraph in section 4.3 into two smaller paragraphs as following: Our findings documented no differences in the daily secretion of cortisol between the clinical groups and the TD children, nevertheless the mean value of the AUCg was significant lower for the ADHD group compared to the control and the ASD group. Even though ADHD has been associated with attenuated HPA axis activity, Angeli et al., have found no differences in CAR and slope cortisol levels between the three ADHD subtypes [77]. Moreover, Anesiadou et al., have found no significant findings regarding the salivary cortisol diurnal secretion in groups of ASD and ADHD children compared to TD [6]. Studies have shown that participation in clinical procedures considered as perceived stressor factor and may lead to an increase of cortisol during the experimental part of the study [78-80]. Similarly, the ADHD and the ASD group showed lower hair cortisol values compared to the TD group. There are plenty of factors that contribute to the age-related cortisol response to psychosocial stress such as sociocognitive maturation. It is known that children with ASD and ADHD have deficits in social and emotional response. Non adaptively social response, lack of theory of mind, a cognitive capacity, characterized by profound difficulties in social interaction and communication prevents an appropriate response to stressful stimuli and environmental demands [81,82].
Activation of the HPA axis coordinates adaption to stress and cortisol is considered essential for normal neurodevelopment. The low levels of cortisol in children with ADHD possibly correlated to Hypo arousal of HPA axis or indicate a delay in daily response secretion [83]. In the study by Pauli-Pott et al., it is highlighted that low hair cortisol in preschool children is a predictor of ADHD developmental disorder and comorbidity with internalizing behavioral problems in school age. [84]. In Ogawa S. et al., study a positive correlation between cognitive functions and hair cortisol was present only in TD children and negative in ASD group. These results underlying the relation between elevated chronic stress hormones in children with ASD and cognitive dysfunctions [85]. Though, other studies suggest that the coordination between the HPA axis and the sympathetic nervous system in generating the physiological adaption of the stress response remains a topic to debate [38]. (lines 375-403)
Comment 14: Some more specific suggestions should be proposed at the end of the Conclusion section concerning the future studies that could be performed in the topic of the present study.
Response 14: We thank the Reviewer for this comment. We added some suggestions for future studies at the end of the Conclusion as following: Nevertheless, in our study have found no significant results regarding salivary cortisol, we suggest future focused studies conducted in other biomarkers of stress system such as salivary α-amylase. (lines 452-456)
Comment 15: There are several typos and grammar/syntax errors throughout the manuscript which should be revised.
Response 15: We thank the Reviewer for this comment. We revised all the text for grammar and syntax errors.
Reviewer 2 Report
Comments and Suggestions for Authors
The manuscript "Functional gastrointestinal symptoms in children with Autism and ADHD: profiles of hair and salivary cortisol, serum leptin concentrations and externalizing/internalizing problems" assesses the presence of FGIDs in children with ASD and ADHD and compares the outcomes with a sample of typically developed (TD) children. The study measured the salivary and hair cortisol concentrations and plasma leptin concentrations in children with ASD, ADHD and children with TD to investigate the potential role of co-morbid internalizing and externalizing behavioral problems, as well as the role of stress-related biomarkers in children with FGIDs and ASD/ADHD. Based on the results, the study reports that the children with Autism and ADHD showed an increased prevalence of FGIDs and internalizing/externalizing problems. Various previous studies have focused mainly on the association between gastrointestinal symptoms (GIS) and developmental regression, language and communication, ASD severity, challenging behavior, comorbid psychopathology, sleep problems, and sensory issues. The submitted manuscript expands the understanding of correlation of GIS with various behavioural problems and subsequent biomarkers.
After going through the manuscript, I have following comments for the authors.
1. The study has included children and adolescent participants (n=120). According to WHO, adolescent is the age group between 10 to 19 years. The adolescents included in the present study were 10-12 years old. How many adolescents were included in the study? Was the number of adolescent participants uniformly distributed in three groups?
2. Adolescence is the age when the individuals experience rapid physical, cognitive and psychosocial growth. Was there any difference in the prevalence of FGIDs in children and adolescents (10-12 years) as well as nales and females?
3. Many abbreviations are used in the manuscript. Please double check that the expanded form of the used abbreviations are mentioned at the point where the abbreviation is used for the first time.
4. Tables need to be improved.
Comments on the Quality of English LanguageMinor grammatical corrections and syntax adjustments needed.
Author Response
Response to Reviewer 2 Comments
We appreciate very much the Reviewers’ interest in our work, for the helpful comments as well as the effort and time put into the review of this manuscript. All comments have been carefully considered and responded point by point.
Comment 1: The study has included children and adolescent participants (n=120). According to WHO, adolescent is the age group between 10 to 19 years. The adolescents included in the present study were 10-12 years old. How many adolescents were included in the study? Was the number of adolescent participants uniformly distributed in three groups?
Response to comment 1: We thank the reviewer for this comment. Twenty-six adolescents (age 10-12) were included in the study and their median age was 11.1 years (IQR=1.32). Sixteen were male (61.5%) with median age 10.2 years (IQR=1.07) and ten were female (38.5%) with median age 11.2 years (IQR=0.97). According to their classification in the 3 study groups, 10 of them belonged to the control group, 3 to the ADHD group and 13 to the ASD group. A significant difference was observed between children and adolescents for their distribution to the 3 study groups (p=0.021), where more adolescents were distributed in the ASD group in contrast to the children where most of them were distributed in the ADHD group.
Comment 2: Adolescence is the age when the individuals experience rapid physical, cognitive and psychosocial growth. Was there any difference in the prevalence of FGIDs in children and adolescents (10-12 years) as well as males and females?
Response to comment 2: We thank the reviewer for this comment. Regarding the prevalence of FGIDs, no statistically significant differences were observed between the subgroups of children and adolescents with FGIDs (p>0.05). Accordingly, no statistically significant differences were found between males and females. However, the number of the children is small, thus we may not found such differences due to the small sample of participants. We added a phrase in the text: “In this study, no significant differences were found between preadolescent and adolescent children with FGIDs, as well as between males and females with FGIDs, possibly due to the small sample” (lines 293-295)
Comment 3: Many abbreviations are used in the manuscript. Please double check that the expanded form of the used abbreviations are mentioned at the point where the abbreviation is used for the first time.
Response to comment 3: We thank the reviewer for this comment. We double checked all used abbreviations that mentioned for the first time.
Comment 4: Tables need to be improved.
Response to comment 4. We thank the reviewer for this comment. We revised our tables.
Reviewer 3 Report
Comments and Suggestions for Authors
Thank you for your contribution to this journal. This is very interesting study to evaluate the Gut-Brain Axis by FGID and cortisol levels.
However, I want to give you some minor comments
1. in line 67, correct "othher" to "other"
2. I wonder why the INT symptoms are more frequent in ADHD with FGID,, not in ASD.
3. This study suffered from the cross-sectional design, you mentioned, and
serum cortisol and hair cortisol is different meaning, eg. serum cortisol present recent stress and hair cortisol is chronic exposure of stress. In terms of stress exposure duration, FGID symptoms, I think, is not always recent stress expression. What is your opinion?
Author Response
Response to Reviewer 3 Comments
We appreciate very much the Reviewers’ interest in our work, for the helpful comments as well as the effort and time put into the review of this manuscript. All comments have been carefully considered and responded point by point.
Comment 1: in line 67, correct "other" to "other"
Response to comment 1: We thank the Reviewer for this comment. We corrected the typo mistake (line 84)
Comment 2: I wonder why the INT symptoms are more frequent in ADHD with FGID,, not in ASD.
Response to comment 2:
We thank the Reviewer for this comment. We also expected a higher rate of internalizing problems in ASD children, however this was not shown, maybe due to the small sample size. A possible explanation may be the complexity of ASD phenotype and the fact that internalizing problems are potentially covered by the core social-communication difficulties in ASD.
We analyze further our consider about INT symptoms in ADHD with FGID compared to ASD.
ASD and ADHD are linked to internalizing problems like anxiety, sadness, social withdrawal, and fearfulness. ADHD traits consist of a stronger and more important predictor of internalizing problems than ASD according to the research. In the majority of studies, ASD is prioritized over ADHD in both research on internalizing problems and clinical practice, though recent research has indicated that comorbidity of internalizing problems is associated with greater social difficulties in ADHD, and it may be difficult to differentiate from ASD symptomology. Furthermore, ADHD traits compared to ASD traits are associated with a lower quality of life, and the lower quality of life is correlated with internalizing problems.
In addition, the executive functions that characterize ADHD and ASD present differentiate neurocognitive pathway. This pathway is still unknown and needs to be studied further. Gut-brain axis is a bidirectional pathway between the gut and the brain, correlated with emotional–behavioural symptoms, which is a common comorbid trait of ADHD. Gut-brain axis is involved in internalizing behavioural problems such as stress, anxiety and depression. So, studies reveal a connection among ADHD traits, internalizing problems and gastrointestinal symptomatology.
References for this response:
- Hargitai LD, Livingston LA, Waldren LH, Robinson R, Jarrold C, Shah P. Attention-deficit hyperactivity disorder traits are a more important predictor of internalising problems than autistic traits. Sci Rep. 2023 Jan 16;13(1):31. doi: 10.1038/s41598-022-26350-4. PMID: 36646771; PMCID: PMC9842645.
- Lee, M.-J.; Lai, H.-C.; Kuo, Y.-L.; Chen, V.C.-H. Association between Gut Microbiota and Emotional-Behavioral Symptoms in Children with AttentionDeficit/Hyperactivity Disorder. J. Pers. Med. 2022, 12, 1634. https:// doi.org/10.3390/jpm12101634
- Esposito D, Bernardi K, Belli A, Gasparri V, Romano S, Terenzi L, Zanatta ME, Iannotti S, Ferrara M. The Hidden Impact of ADHD Symptoms in Preschool Children with Autism: Is There a Link to Somatic and Sleep Disorders? Behav Sci (Basel). 2024 Mar 6;14(3):211. doi: 10.3390/bs14030211. PMID: 38540514; PMCID: PMC10967894.
Comment 3. This study suffered from the cross-sectional design, you mentioned, and serum cortisol and hair cortisol is different meaning, eg. serum cortisol present recent stress and hair cortisol is chronic exposure of stress. In terms of stress exposure duration, FGID symptoms, I think, is not always recent stress expression. What is your opinion?
Response to comment 3: We thank the Reviewer for this comment. In our study we assessed salivary, hair cortisol and leptin serum as stress biomarkers. We hypothesized that hair cortisol, as a chronic stress biomarker (last 3 months), might be predictive of FGIDs (that reflect symptoms in the last two months ), but salivary cortisol, as a current stress biomarker, might also offer valuable information.
This hypothesis was not confirmed probably due to the small sample.
Studies report that both acute and chronic stress may influence gastrointestinal tract and they are considered risk factors for the development and maintenance of FGID. Activation of stress system, internalizing/externalizing comorbidity and gastrointestinal symptoms appear in the sense of a vicious circle. Chronic stress can induce somatic symptoms such as FGIDs by altering the intestinal motility and secretion, changing the composition of gut microbiota and exacerbate inflammatory processes. Acute stress increases the intestinal secretion and gut permeability which may lead to acute gastrointestinal symptoms such as diarrhea. The kind of stressors and the exposure time to a stressor stimuli have different physiological consequences in the gastrointestinal tract.
References for this response:
- Leigh, SJ.; Uhlig, F.; Wilmes, L.; Sanchez-Diaz, P.; Gheorghe, C.;. Goodson, MS.; Kelley-Loughnane, N.; Hyland, NP.; Cryan, J.; Clarke, G. The impact of acute and chronic stress on gastrointestinal physiology and function: a microbiota–gut–brain axis perspective. The Journal of Physiology published by John Wiley & Sons Ltd on behalf of The Physiological Society. 2023; https://doi.org/10.1113/JP28195
- Procházková, N., Falony, G., Dragsted, L. O., Licht, T. R., Raes,J., & Roager, H. M. (2023). Advancing human gut micro-biota research by considering gut transit time.Gut,72(1),180
- McEwen, B. S. (2017). Neurobiological and systemic effectsof chronic stress.Chronic Stress (Thousand Oaks),1,2470547017692328
- Wu JC. Psychological Co-morbidity in Functional Gastrointestinal Disorders: Epidemiology, Mechanisms and Management. J Neurogastroenterol Motil. 2012 Jan;18(1):13-8. doi: 10.5056/jnm.2012.18.1.13. Epub 2012 Jan 16. PMID: 22323984; PMCID: PMC3271249.